

# Diversity and structure of the root-associated bacterial microbiomes of four mangrove tree species, revealed by high-throughput sequencing

Jinlei Sui[1,*], Xiaowen He[1,*], Guohui Yi[1], Limin Zhou[1], Shunqing Liu[1], Qianqian Chen[1], Xiaohu Xiao[2] and Jinyan Wu[1]

[1] Public Research Center, Hainan Medical College, Haikou, China
[2] Rubber Research Institute, Chinese Academy of Tropical Agricultural Sciences, Haikou, China
[*] These authors contributed equally to this work.

## ABSTRACT

**Background**. Root-associated microbes of the mangrove trees play important roles in protecting and maintaining mangrove ecosystems. At present, most of our understanding of mangrove root-related microbial diversity is obtained from specific mangrove species in selected geographic regions. Relatively little is known about the composition of the bacterial microbiota existing in disparate mangrove species microenvironments, particularly the relationship among different mangrove species in tropical environments.

**Methods**. We collected the root, rhizosphere soil, and non-rhizosphere soil of four mangrove trees (*Acanthus ilicifolius*, *Bruguiera gymnorrhiza*, *Clerodendrum inerme*, and *Lumnitzera racemosa*) and detected the 16S rRNA gene by a conventional PCR. We performed high throughput sequencing using Illumina Novaseq 6000 platform (2 × 250 paired ends) to investigate the bacterial communities related with the different mangrove species.

**Results**. We analyzed the bacterial diversity and composition related to the diverse ecological niches of mangrove species. Our data confirmed distinct distribution patterns of bacterial communities in the three rhizocompartments of the four mangrove species. Microbiome composition varied with compartments and host mangrove species. The bacterial communities between the endosphere and the other two compartments were distinctly diverse independent of mangrove species. The large degree of overlap in critical community members of the same rhizocompartment across distinct mangrove species was found at the phylum level. Furthermore, this is the first report of *Acidothermus* found in mangrove environments. In conclusion, understanding the complicated host-microbe associations in different mangrove species could lay the foundation for the exploitation of the microbial resource and the production of secondary metabolites.

Corresponding authors
Xiaohu Xiao, xiaofeihu_011588@126.com
Jinyan Wu, hy0303003@hainmc.edu.cn

## INTRODUCTION

Located at the transition zone of terrestrial and marine systems, mangrove forests have a rich biodiversity of animals, plants, and microbiota. Dongzhaigang mangrove wetland, a tropical mangrove forest, occupies 45 percent of the mangrove wetland and is the largest contiguous mangrove ecosystem in Hainan, China. Meanwhile, it is a nature reserve with the richest mangrove tree species, accounting for exceeding 60 percent of the mangrove tree species in China (*Liu, Peng & Li, 2012*). The unique adaptation of mangrove species to intertidal zones subjects them to highly variable environmental conditions, resulting in the establishment of diverse bacterial communities that characterize the mangrove ecosystem (*Thatoi et al., 2013*).

Mangrove plants and their endophytes have garnered significant attention as natural sources of novel bioactive compounds, attracting the interest of numerous scientists in recent years (*Ancheeva, Daletos & Proksch, 2018*). Among the various mangrove plants, approximately 20 species have been empirically proven to possess medicinal value (*Nabeelah Bibi et al., 2019*). Previous research has demonstrated that the extracts from different tissues of mangrove plants *Acanthus ilicifolius* Linn. (abbr. as *A. ilicifolius* below), *Bruguiera gymnorrhiza* (Linn.) Savigny (abbr. as *B. gymnorrhiza* below), *Clerodendrum inerme* (Linn.) Gaertn. (abbr. as *C. inerme* below), and *Lumnitzera racemosa* Willd. (abbr. as *L. racemosa* below) or metabolites of their endosphere have been applied to the cure of various diseases (*Fu, Wang & Shao, 2009*; *Mohan & Mishra, 2010*; *Thuy et al., 2019*; *Wu et al., 2016*; *Yan et al., 2020*).

In recent years, the scarcity of resources related to wild and rare medicinal plants has driven a growing number of researchers to focus on their endophytic and rhizosphere microorganisms (*Aghdam & Brown, 2021*). These plant-associated microorganisms have emerged as crucial source of alternative medicinal plant resources and new active substances due to their unique properties (*Gomes et al., 2010*). Endophytic bacteria, which inhabit plants for all or part of their life cycle without inducing any diseases in the plants, have garnered considerable interest (*Compant et al., 2021*). Rhizosphere bacteria, a subset of soil bacteria, primarily inhabit the rhizosphere and are initially attracted to plant roots through the exudation of root substances into the surrounding soil (*Yu & Hochholdinger, 2018*). The investigation of endophytic bacteria in medicinal plants began with the discovery of the endophytic fungus *Taxomyces andreanae*, which produces taxol and taxane, in *Taxus brevifolia* (*Stierle, Strobel & Stierle, 1993*). Subsequent studies have increasingly demonstrated that endophytes can produce similar or identical active compounds to those found in host plants, owing to their evolutionary process over time (*Aghdam & Brown, 2021*). Root endophytes, primarily derived from soil, share a significant relationship with rhizosphere microorganisms and serve as effective supplements to root endophytic bacteria (*Hamonts et al., 2018*). Therefore, studying endophytic and rhizosphere bacteria in conjunction allows for a more comprehensive exploration of endophytic bacteria resources and establishes a foundation for investigating a wider range of bioactive substances through comparative analyses of their interactions.
Plants grown naturally in the soil develop tight relationships with soil microbiota (*Zhuang et al., 2020*). The root microhabitat is usually divided into three rhizocompartments (*i.e.,* endosphere, rhizosphere, and non-rhizosphere), which refer to compartments surrounding the plant root system (*Edwards et al., 2015*). The endosphere comprises all inner root tissues, which is inhabited by microbes (*Chen et al., 2021*). The rhizosphere, a narrow soil zone in direct proximity to the root system, is highly influenced by the root system itself (*Yu & Hochholdinger, 2018*). The non-rhizosphere soil separated from the root by shaking was used as a control to differentiate plant effects from general edaphic factors. Distinct distribution patterns for bacterial communities were found in different compartments, which were related to niche differentiation along the root compartments (*Edwards et al., 2015*; *Zhuang et al., 2020*). The rhizosphere could emit root exudates that selectively enriched specific microbial populations; however, these exudates were found to exert only marginal impacts on microbes in the non-rhizosphere soil, which makes the abundance and species of rhizosphere microorganisms differ to some extent from those of non-rhizosphere soils (*Thatoi et al., 2013*; *Zhuang et al., 2020*). Differences between microbial communities in the root-related compartments from distinct plants are observed at different taxonomic levels and are associated with the root-zone environment (*Ofek-Lalzar et al., 2014*). Plant microbiota plays a key role in protecting and sustaining plant ecosystems. On the one hand, the endophytic microbiome is in contact with the plant's internal environment and absorbs nutrients from the host tissues, which is closely associated with their growth and development. On the other hand, microorganisms colonizing host plants can produce certain chemicals, which may affect the synthesis and accumulation of host secondary metabolites, and may also be an important source of plant medicinal ingredients (*Korenblum et al., 2020*).

Similar to other medicinal plants, mangrove medicinal plants have also adapted to utilize their tight association for their great benefit (*Thatoi et al., 2013*). Highly diverse microbiomes have been proven to live and function in the roots of mangrove species (*McKee, 1993*; *Srikanth, Lum & Chen, 2016*). Increasing evidence has been provided to support the significance of root-related microorganisms for the growth, development, and metabolism of mangrove trees (*Liu et al., 2020*; *Zhuang et al., 2020*).

Plant root microbiotas vary by agrotype and plant genotype and have a significant influence on plant health and ecosystem stability (*Bulgarelli et al., 2012*; *Haichar et al., 2008*; *Ofek-Lalzar et al., 2014*). A previous study demonstrated that microbiome assembly along the soil-plant continuum is shaped predominantly by compartment niches and host species for crops (*Xiong et al., 2021*). So far, only a few studies have been conducted on the microbiome of different mangrove species. The distributions of rhizosphere bacteria from three mangrove species in Beilun Estuary, South China showed that mangrove tree species strongly influenced the bacterial community, but only the composition of rhizosphere bacteria was analyzed (*Wu et al., 2016*). Similarly, the bacterial diversity and community structure in the rhizospheres of four mangrove species (*Sonneratia alba, Rhizophora mucronata, Ceriops tagal* and *Avicennia marina*) from Mida Creek and Gazi Bay, Kenya was evaluated, which data for other rhizocompartments were also lacking (*Muwawa et al., 2021*). Therefore, a comprehensive and systematic study on the structure of mangrove

rhizomes in different compartments of different mangrove species is lacking. Based on the universality of the distribution, the soil types, and their medicinal value, four mangrove species (*A. ilicifolius*, *B. gymnorrhiza*, *C. inerme*, and *L. racemosa*) were chosen for this study (*Fu, Wang & Shao, 2009*; *Mohan & Mishra, 2010*; *Thuy et al., 2019*; *Wu et al., 2016*; *Yan et al., 2020*).

In our present work, we hypothesized that root-associated microbiome composition would vary with rhizocompartments and host mangrove species and have unique characteristics. By employing a replicated experimental design, we examined the composition and variability of bacterial microbiomes within three distinct rhizocompartments (non-rhizosphere, rhizosphere, and endosphere) across four significant and diverse mangrove tree species (*A. ilicifolius*, *B. gymnorrhiza*, *C. inerme*, and *L. racemosa*) in a tropical mangrove nature reserve. Our findings shed new light on the intricate microbial associations within the roots of mangrove trees situated in tropical island environments and contribute to a deeper understanding of root-associated microbial communities, thus expanding our knowledge of mangrove ecosystems and their ecological dynamics.

## MATERIAL AND METHODS

### Plant materials and sampling

The sampling site was located at Dongzhaigang National Natural Reserve (110°32′110° 37′E; 19°95′20°1′N), Haikou City, Hainan Province, China (Fig. 1), which harbors mangrove communities with the highest abundance in China. Four different mangrove tree species, *A. ilicifolius*, *B. gymnorrhiza*, *C. inerme*, and *L. racemosa* were collected on May 3, 2021. For each mangrove species, five biological replicates, each comprising five pooled subsamples were collected, as well as their surrounding soils. Compartments surrounding the plant root system are known as rhizocompartments, which comprise the endosphere (*i.e.,* all inner root tissues), the rhizosphere soil (*i.e.,* the soil immediately surrounding the root) and the non-rhizosphere soil (*i.e.,* the soil not immediately surrounding the root). These distinct compartments collectively influence various physiological and ecological processes that shape the overall functioning of plants. To disclose the bacterial microbiome composition across three rhizocompartments of four mangrove species, the samples we collected from each species were also divided into three compartments: endosphere (R), rhizosphere soil (S), and non-rhizosphere soil (N). They were processed within 12 h after sampling. The soil around the root not adhering directly to the roots was defined as the non-rhizosphere soil fraction. The soil (about one mm thickness) is attached to the root by washing with sterile Phosphate Buffered Saline (PBS) solution was collected as a rhizosphere fraction. To obtain the endosphere biomass, root systems were washed by constant shaking in TE buffer containing 0.1% Triton X-100, followed by washing consecutively in 75% ethanol and 2.5% Sodium hypochlorite to further rinse the root surfaces, and then washed five times (3 min each) with sterile distilled water.

### DNA extraction, PCR amplification, and high-throughput sequencing

The root was homogenized in advance by bead beating for 1 min before the DNA isolation (Mini Beadbeater, Jingxin, and Shanghai). Root, rhizosphere, and non-rhizosphere soil

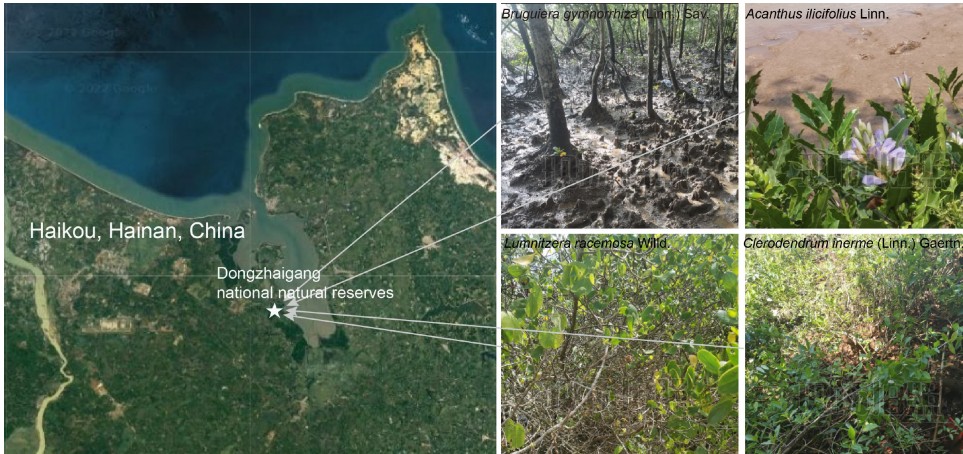

**Figure 1** **Location and growing environments of the four mangrove populations. The site was in Dongzhaigang national natural reserve and its surrounding area.** The non-rhizosphere, rhizosphere, and root of four mangrove species (*B. gymnorrhiza*, *A. ilicifolius*, *C. inerme,* and *L. racemosa*) were harvested in summer 2021. Base map sourced from Google Maps (Map data ©2023 Google).

(approximately 0.5 g) with five biological replicates used for DNA isolation. The total DNA for each sample was then isolated using the protocol of the MoBio PowerSoil® DNA isolation kit (Mo Bio Laboratories Inc, USA, catalog No. 12888-100) with slight modifications. Briefly, the kit mainly utilized a bind/wash/elute workflow. The test sample was treated with lysis buffer (Solution C1). Following a brief centrifugation at $10,000 \times g$ for 30 s, the resulting supernatant was carefully collected and transferred to a pristine two mL Collection Tube. To effectively eliminate impurities, repeated additions of Solution C2 were employed, with subsequent supernatant collection achieved through centrifugation. To ensure a robust binding of the DNA to the Spin Filter, Solution C4, characterized by high salinity, was introduced and thoroughly mixed with the supernatant. Subsequently, the resulting solution, comprising approximately 675 µL, was loaded onto the Spin Filter. This was accompanied by centrifugation at 10,000 x g for 1 min, with the process repeated three times to enhance the binding efficacy. Following this binding step, the Spin Filter underwent successive washes using Solution C5 (500 µL). Ultimately, the elution of the DNA of interest was achieved through two rounds of centrifugation at 10,000 x g for 30 s each, utilizing solution C6 (50 µL). The quality of DNA was determined by OD(260/280) and OD(260/230) ratios through a NanoDrop spectrophotometer.

The 16S rRNA genes (V3 −V4 region) were amplified using primer pairs (forward primer, 5′-ACTCCTACGGGAGGCAGCA-3′; reverse primer, 5 ′- GGACTACHVGGG TWTCTAAT-3′) as described previously (*Wang, Chen & Zhang, 2017*). The polymerase chain reaction (PCR) was carried out in a reaction volume of 50 µL, containing 5 µL 10×Pfu Buffer, 2 µL 2.5 mM dNTP Mixture, 2 µL each primer set (10 µM), 10 µL 5xGC Enhancer 1 µL Pfu (2.5 U/ µL), 50 ng template DNA. PCR was performed according to a previous study (*Zhuang et al., 2020*). Before amplicon sequencing, PCR products of all samples were detected by running agarose gel and quantified through a NanoDrop

spectrophotometer. The high-quality 16S rRNA amplicons with the same concentrations from each sample were pooled and then sent for sequencing using the Illumina Novaseq 6000 platform by generating PE reads in targeted regions (Biomarker, Beijing, China).

## Amplicon-based microbiome data analysis

Valid amplicon sequences were obtained from the verified libraries using the Illumina Novaseq 6000 system. These sequences were then transformed into raw reads through a base-calling algorithm, and assembly was performed using FLASH (Version 1.2.11). To generate high-quality amplicon sequences, a series of tools including Trimmomatic (Version 0.33), Cutadapt (version 1.9.1), Usearch (Version 10.0), and UCHIME (Version 4.2) were employed. During the filtering process with Trimmomatic, sequences were trimmed using a 50-bp moving window approach, while maintaining a quality threshold Q-score of 20. Cutadapt was utilized to identify and remove primer sequences, employing a maximum mismatch ratio of 0.2 and a minimum coverage of 80%. For the assembly of paired-end (PE) reads, Usearch was utilized, requiring a minimum overlap length of 10 bp, a minimum similarity within the overlapping region of 90%, and a maximum accepted mismatch of 5 bp. Subsequently, UCHIME was utilized to eliminate chimeric sequences, using a similar threshold of greater than 80% compared to the query sequence. The resulting high-quality reads, obtained through the aforementioned steps, were used for subsequent analyses. The operational taxonomic units (OTU) as the final valid reads were generated by 97.0% sequence similarity with Usearch software (Version 10.0) and the conservative threshold for OTU filtration is 0.005%. The annotation of feature sequences was performed using a bayesian classifier compared with the SILVA reference sequences (Release 132, http://www.arb-silva.de) (*Quast et al., 2013*). The sequences matching the reference sequences annotated as "Chloroplast" and "Mitochondria" were removed from the datasets.

Bioinformatics analysis was carried out using the cloud platform BMKCloud (http://www.biocloud.net). QIIME2 (https://qiime2.org/) was used for evaluating the abundance of different species in samples, and the histogram based on frequency distributions at different taxonomic levels was performed using an R package. Subsequently, the $\alpha$-diversity index of samples was also obtained by the QIIME software, using Student's $t$-test to analyze the bacterial diversity and richness of each mangrove species. The ACE and Chao1 indices were used for estimating the bacterial abundance, while the Simpson and Shannon indices were been applied to estimate the bacterial diversity. Principal coordinates analysis (PCoA) was performed to test the distinction in $\beta$-diversity through the https://en.wikipedia.org/wiki/UniFrac (Binary Jaccard and Unweighted Unifrac) algorithms to measure the distance (OTU level). Non-metric multidimensional scaling (NMDS) obtained from the Binary-Jaccard distance was applied to investigate the difference between different groups. Biomarkers (significantly different OTUs) between different groups were filtered out by LEfSe (Line Discriminant Analysis (LDA) Effect Size) (*Quast et al., 2013*). The taxonomic ranks were from phylum to species and the LDA threshold was set to 4.0. A network diagram is used to reveal the correlation. Spearman rank correlation analysis was applied to evaluate the associations and variations in abundant species ($|r| > 0.9$;

$p < 0.05$) and then used for building the network. To predict potential functions based on the 16S rRNA gene amplicon data, the Functional Annotation of Prokaryotic Taxa (FAPROTAX Version 1.2.6) database (http://www.loucalab.com/archive/FAPROTAX) was utilized (*Louca, Parfrey & Doebeli, 2016*). The FAPROTAX database was used to predict possible functions based on the data of 16S rRNA gene amplicons. FAPROTAX comes with a versatile script (collapse_table.py) for converting taxonomic microbial community profiles (*e.g.*, in the form of an OTU table) into putative functional profiles based on the published and verified culturable bacteria literature. The complete database for FAPROTAX includes over 7600 functional annotations covering over 4600 taxa, and over 80 functions, being freely available at http://www.loucalab.com/archive/FAPROTAX. It maps prokaryotic taxa (*e.g.*, genera or species) to metabolic or other ecologically relevant functions (*e.g.*, nitrification, denitrification or fermentation).

## RESULTS

### General analysis of the sequencing data

The sequencing of the bacterial 16S rRNA amplicons of four distinct mangrove species, *A. ilicifolius*, *B. gymnorrhiza*, *C. inerme,* and *L. racemosa* yielded 4,314,420 PE reads in total from 60 samples. After quality filtering and read merging, 3,244,252 high-quality sequences in total were obtained (Table S1). With a threshold of 97% sequence identity and deleting low-abundance OTUs, 2636 OTUs were detected (Table S1). Among them, 2,119 bacterial OTUs were in common among the four distinct mangrove species in all samples (Fig. 2A), and 1,608 bacterial OTUs were shared among the three different rhizocompartments (Fig. 2B). Nine unique OTUs of *C. inerme* were found and the number was much higher than the other three mangrove species (Fig. 2A), this may be related to the fact that the sampling site of *C. inerme* was on the shore, and the soil was dry, while the soil of other three mangrove species was muddy. Meanwhile, more unique OTUs were observed in the endosphere than that in the other two compartments (Fig. 2B). This is probably because the endophytic microorganisms in the root are affected not only by the soil environment but also by physiological conditions, growth stages, and metabolites of the plant. The sample-based Shannon curves and OTU species accumulation curves of bacterial OTUs nearly achieved the saturation state (Fig. S1), and Good's coverage obtained for all the test samples was over 99 percent (Table S2), manifesting that the sequencing depth was enough to cover most bacterial ranks in all the tested samples.

### Composition of bacterial communities

We compared the composition analyses of the root-associated compartments in four different mangrove species, based on the 16S rRNA amplicon dataset. The total OTUs in all samples were classified and designated into 34 phyla, 94 classes, 232 orders, 385 families, and 646 genera.

The abundant bacteria at the taxonomic ranks (phylum, class, and genus) among three different compartments of four different mangrove species are shown in Figs. 3A–3C. At the phylum level, the dominant bacteria in all compartments of all samples (endosphere, rhizosphere soil, and non-rhizosphere soil) was Proteobacteria, accounting

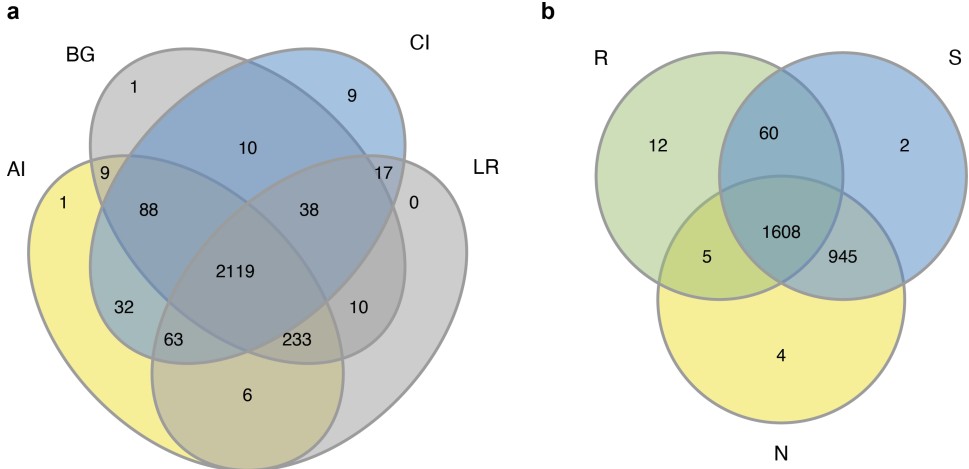

**Figure 2** Venn diagrams of the bacterial operational taxonomic units (OTUs) in four different mangrove species (A) or among three different compartments (B). AI, BG, CI, and LR refer to *A. ilicifolius*, *B. gymnorrhiza*, *C. inerme*, and *L. racemosa*, respectively; N, S, and R refer to non-rhizosphere soil, rhizosphere soil, and endosphere, respectively.

for 62.60%, 47.99%, and 45.53%, respectively, while the second dominant bacteria in each compartment was different. As the core compartment, the endosphere was distinctive. The second most common phylum was Actinobacteria (20.57%) in the root endosphere compartment, while it was Chloroflexi in the other two compartments (Fig. 3A). At the class level, Gammaproteobacteria was the most abundant which was 58.14% in endosphere, 23.04% in rhizosphere soil, and 19.21% in non-rhizosphere soil, respectively. Actinobacteria (18.64%) was the second predominant class in the endosphere compartment. Alphaproteobacteria was the second dominant class in other compartments except in roots, the relative abundances were 12.85% in the rhizosphere and 16.67% in the non-rhizosphere respectively (Fig. 3B). At the genus classification level, we found significant distinctions across three root-associated rhizocompartments. The root endosphere had a significantly higher proportion of *Marinomonas, Vibrio*, and *Acidothermus* than the other two rhizocompartments, whereas uncultured members of Gammaproteobacteria, Gaiellales, and Rhodobacteraceae were almost depleted in the endosphere (Fig. 3C).

The abundant bacteria at the classification levels (phylum, class, and genus) in four different mangrove species are shown in Figs. 3D–3F. The phylum-level similarity of the bacterial community was found in different mangrove species, although the proportion was different. *C. inerme* had a larger proportion of Actinobacteria, Acidobactera, and Gemmatimonadetes, while *C. inerme* had a smaller proportion of Proteobacteria. However, in *B. gymnorrhiza*, the bacteria with the lower abundance belonged to Actinobacteria. At the class level, the bacterial composition of *C. inerme* was quite different from the other three species, with a smaller proportion of Gammaproteobacteria and Deltaproteobacteria. In the top 15 abundant classes, most of the taxa had a similar ratio. Notable genus-level differences among the four mangrove species were reported in this study. In *A. ilicifolius* and *B. gymnorrhiza*, the predominant genus was *Vibrio*, while the dominant bacteria

Peer J

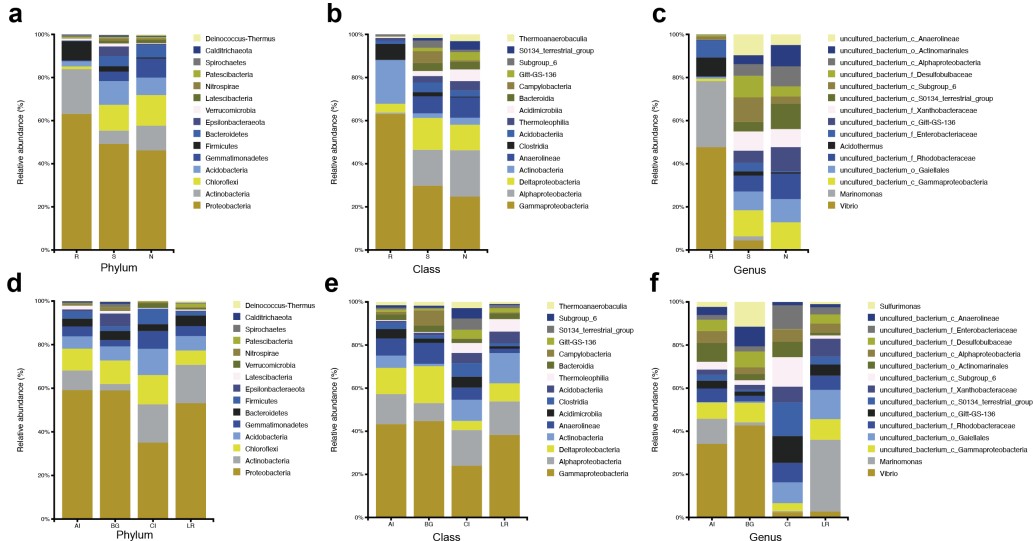

**Figure 3** Microbial community structure in four natural mangrove populations. (A–C) Relative abundances of the bacteria at the levels of phylum, class, and genus of different compartments (R, S, or N) in all mangrove populations. (D–F) Relative abundances of the bacteria at the levels of phylum, class, and genus of different mangrove species (AI, BG, CI, or LR) in all compartments. Only the top 15 dominant taxa were shown in this figure.

in *L. racemosa* was *Marinomonas*. In *C. inerme*, an uncultured terrestrial bacterium was dominant, which was consistent with the root's surroundings. The mangrove tree *C. inerme* grows in drier, higher land, and is submerged by sea water for less time, so its root-associated microorganisms are characteristic of land plants.

At the phylum level, Actinobacteria was the second common phylum in the root endosphere compartments of four different mangrove species except for *B. gymnorrhiza*, (Fig. 4A). *B. gymnorrhiza* trees selected in our study were soaked in seawater and mud all year round, resulting in an oxygen shortage of the roots, thus affecting the growth and reproduction of aerobic actinomycetes. However, at the genus level, the bacterial composition of the same compartment in different mangrove species is quite different from each other (Fig. 4B).

In each compartment of each mangrove species, the microbial composition and structure are similar to that of all samples with some differences. At the phylum level, in all compartments of *C. inerme* and *L. racemosa*, Actinobacteria accounts for a greater proportion, whereas it is depleted in *A. ilicifolius* and *B. gymnorrhiza* (Figs. 4C–4D). In the four different mangrove plants, the microbial composition in the roots of *A. ilicifolius*, *B. gymnorrhiza*, and *L. racemosa* was significantly different from that in the other two compartments. In *C. inerme*, the structure of the bacterial community between the root and other compartments are the most similar, which may be because the *C. inerme* trees we selected grow in a nutrient-poor shore environment, which is less affected by seawater, and thus the connection between the root and the rhizosphere was more closely. At the genus level, the differences between the three compartments were more obvious. Although
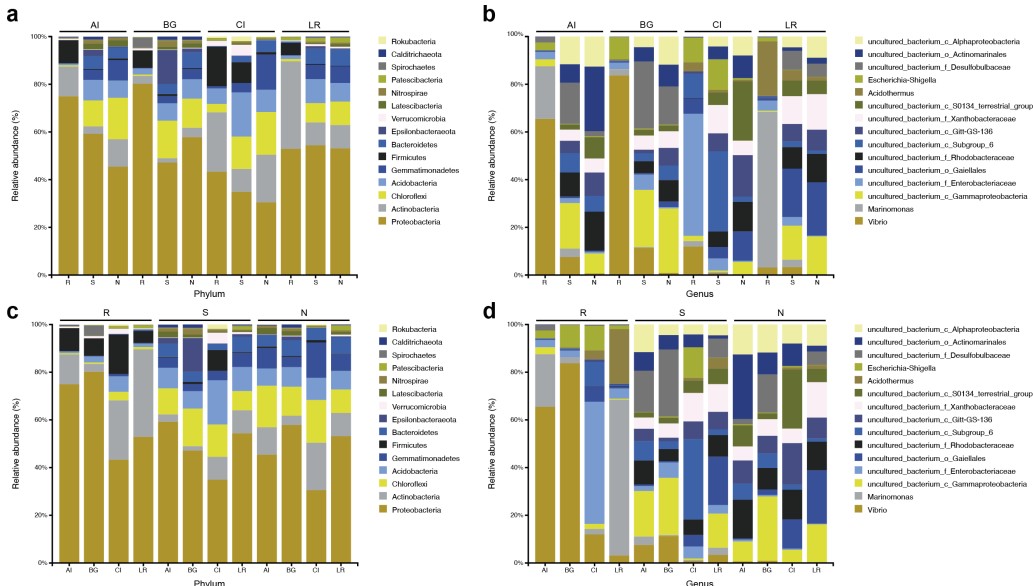

**Figure 4 Microbial community structure in four different natural mangrove populations or three different compartments.** (A–B) Relative abundance of the bacteria of three mangrove root-associated compartments (R, S, and N) in four different mangrove species (AI, BG, CI, or LR) at phylum and genus level, respectively. (C–D) Relative abundance of the bacteria of four different mangrove species (AI, BG, CI, or LR) in three mangrove root-associated compartments (R, S, and N) at phylum and genus levels, respectively. Only the top 15 dominant taxa were shown in this figure.

there are similar groups at the phylum level, the predominant bacteria are quite different at the genus level, which may be relevant to the host and environmental differences between them.

## Comparisons of bacterial communities among distinct root-associated compartments

The richness and diversity of microbial communities among different root-associated compartments in four mangrove species were estimated by $\alpha$-diversity indices, including ACE, Chao1, Shannon, and Simpson (Table S2). In terms of $\alpha$-diversity, there was no significant difference between rhizosphere and non-rhizosphere microbial diversity whether ACE, Chao1, Shannon, or Simpson indices were selected. However, the diversity between endosphere and the other two compartments were remarkably different ($P < 0.05$) (Figs. 5A–5D). In general, the diversity in the root endosphere compartment was much lower than that in the other two compartments. This phenomenon was consistent with previous findings, illustrating that the abundance of microbial communities in the rhizosphere was higher compared to the endosphere under a high-nutrient soil environment (*Orozco-Mosqueda et al., 2018*).

$\alpha$-diversity between rhizospheric compartments illustrated a decreasing gradient in bacterial diversity from the rhizosphere to the endosphere independent of mangrove species (Figs. 5A–5D and Table S2). Bacterial communities of the endosphere had the lowest $\alpha$-diversity, while that of the rhizosphere had the highest $\alpha$-diversity, although

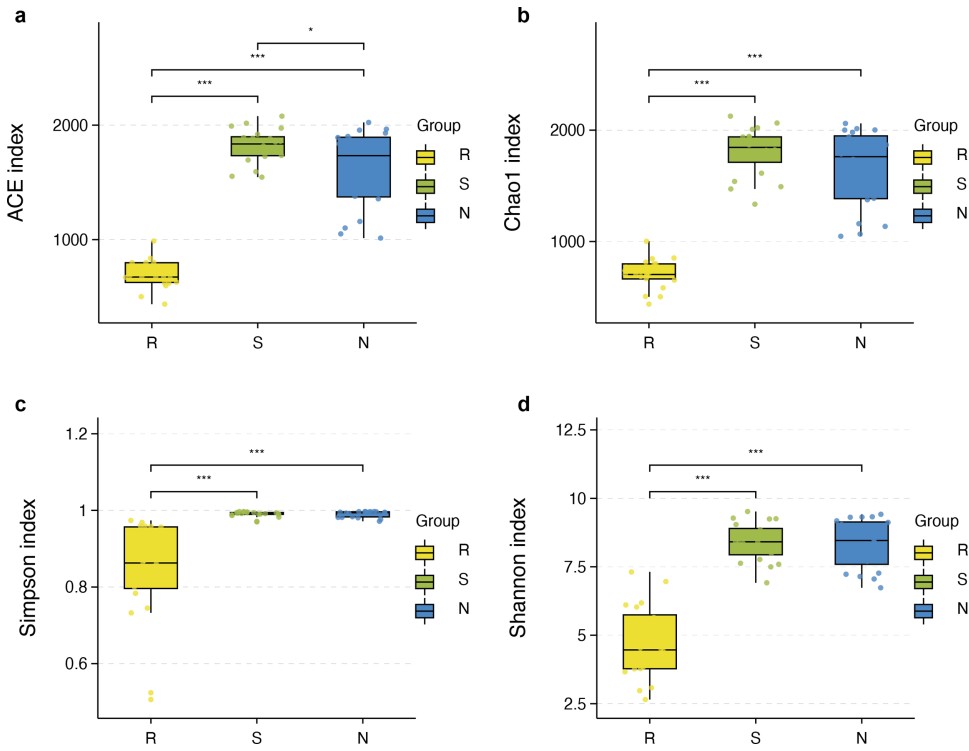

**Figure 5** **Boxplot of alpha diversity metrics crosses three root-associated compartments.** (A–D) depict ACE, Chao1, Simpson, and Shannon index of root-associated compartments in all samples, respectively. The horizontal bars within the boxes represent the median. The tops and bottoms of boxes represent the 75th and 25th quartiles, respectively. The upper and lower whiskers extend 1.5 × the interquartile range from the upper edge and lower edge of the box, respectively. All samples are plotted as individual points.

$\alpha$-diversity indices were similar between the two exterior rhizocompartments. As shown in Fig. S2 $\alpha$-diversity between each mangrove species revealed no obvious difference, although the structure of bacterial communities was different.

The microbial community structure comparisons among different root-associated compartments in four mangrove species were performed by PCoA using the Binary Jaccard and Unweighted Unifrac algorithms (Figs. 6A–6B). In this type of analysis, the distance shows the similarity of bacterial populations. Therefore, samples with high similarity are inclined to cluster together. As shown in Figs. 6A–6B, bacterial communities of the two exterior rhizocompartments were located closer and were more similar to each other than those of the endosphere. Remarkable differences were obtained between the root endosphere and the other two compartments along the PC1 axis (*P* value <0.05), so the $\beta$-diversity of bacteria in the endosphere compartment exhibits a uniqueness from the other two compartments. In addition, the pattern of separation is in accord with a gradient of bacterial populations from the interior of the root into the exterior of the root. The $\beta$-diversity of bacteria among four mangrove species exhibits no significant difference (Figs. 6C–6D).

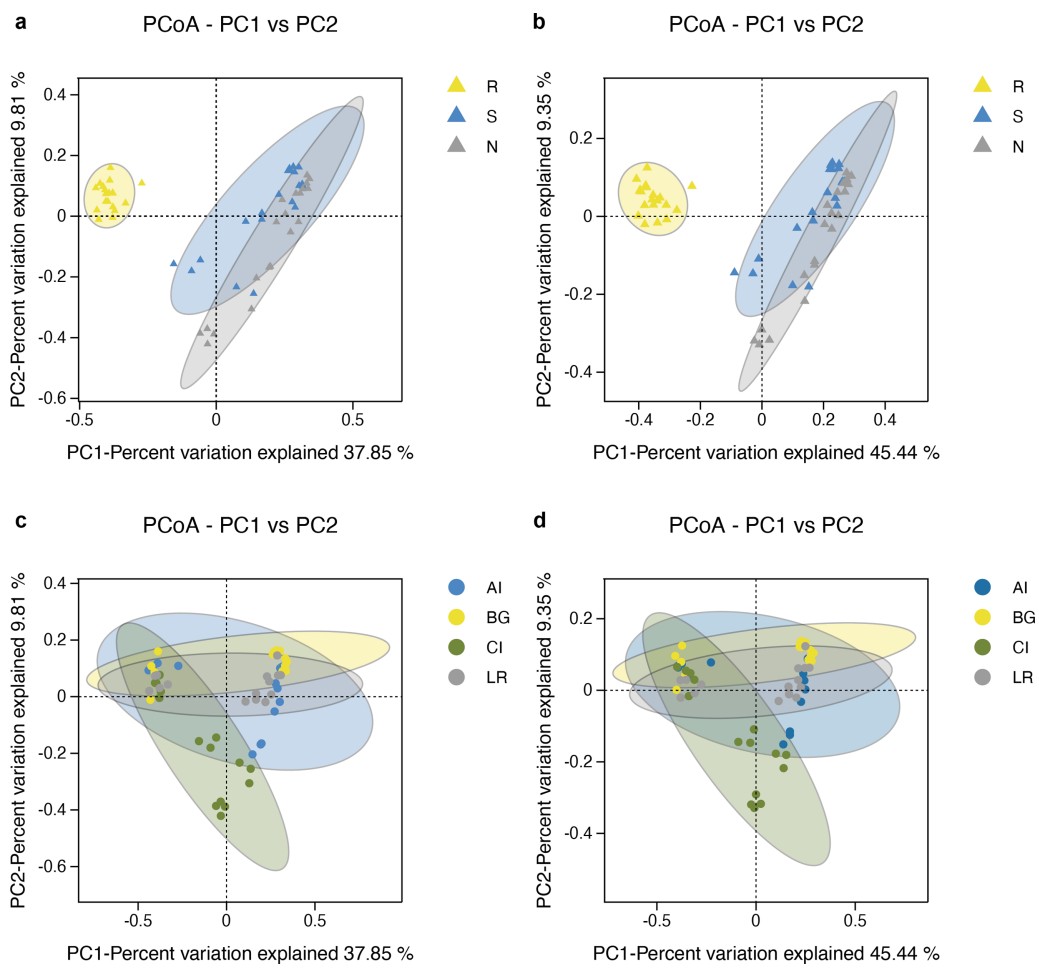

**Figure 6** Principal coordinates analysis (PCoA) of bacterial communities and the 95% confidence ellipses are shown around the samples and grouped based on three root-associated compartments (R, S and N) (A, B) or four mangrove species (C, D). (A, C) PCoA using the BJ (Binary-Jaccard) metric; (B, D) PCoA using the UUF (Unweighted UniFrac) metric.

As shown in Fig. 7A, the UPGMA clustering tree and histogram of species composition based on UniFrac distance showed that the root endosphere of four mangrove species was clustered together, while the samples from the other two compartments were clustered together separately, but there were some overlaps between these two compartments samples. These results were consistent with PCoA (Figs. 6A–6B).

Non-metric multidimensional scaling analysis (NMDS) using the Binary-Jaccard algorithm was carried out to compare the bacterial community structure in different compartments or different mangrove species (Figs. 7B–7C). NMDS results in Fig. 7B well revealed the difference in bacterial community structure in root compartments ($R^2 = 0.36$, $P = 0.001$). The bacterial communities in the endosphere compartment differed significantly as compared to the other two compartments, suggesting the selective entry and exit of bacteria communities on the surface of the root, while the differences in

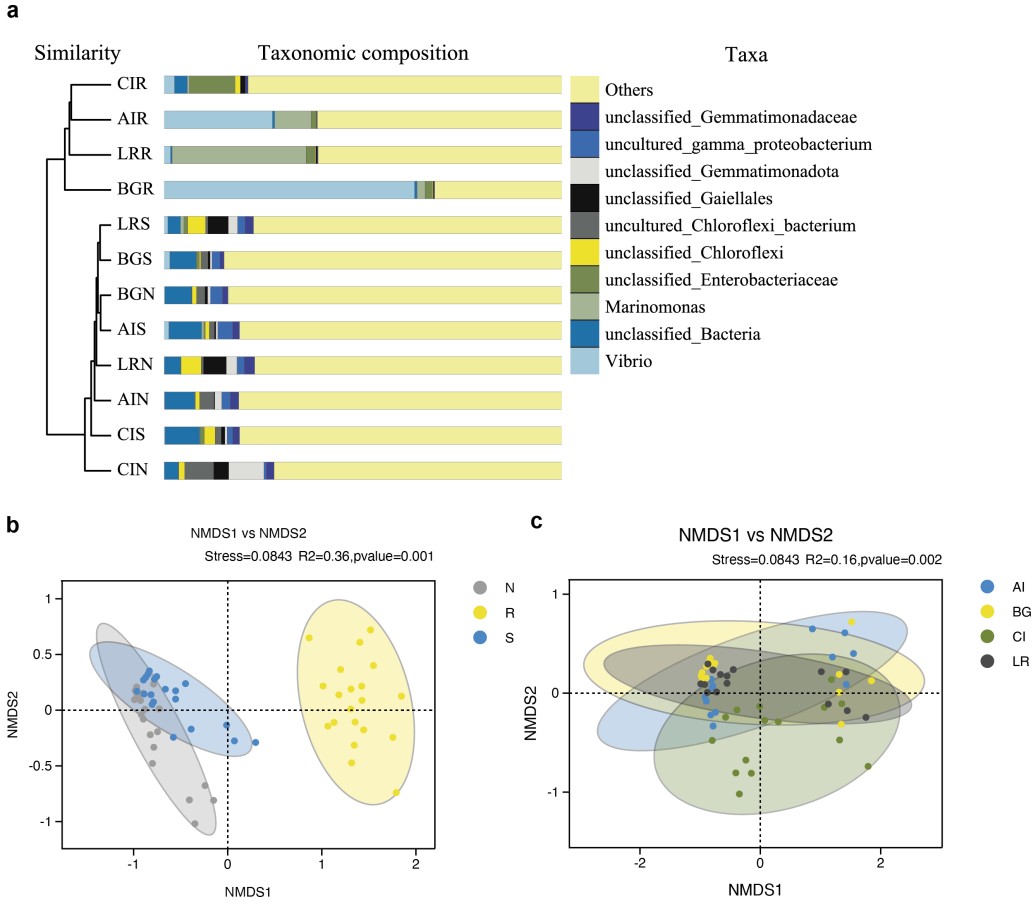

**Figure 7 Unweighted pair-group analysis (UPGMA) and non-metric multidimensional scaling analysis (NMDS) of bacterial community structure at the OTUs level in different compartments or in different mangrove species.** (A) UPGMA based on UniFrac distance for bacterial communities in three root-associated compartments. (B) NMDS in different compartments. (C) NMDS in different mangrove species. The circles in figure B–C indicate each sample separately, different colors represent different groups, and the distance between circles indicates the degree of difference. Stress value less than 0.2 means that NMDS analysis is reliable, and the closer samples are on the coordinate figure, the higher similarity.

bacterial communities among different mangrove species were not significant ($R^2 = 0.16$, $P = 0.002$).

## Differential analysis between different compartments and different mangrove species

LEfSe analysis was applied to seek the biomarkers with statistically significant differences between the groups. LEfse analysis sought out significantly different genera abundant in three root-associated compartments or four distinct mangrove species ($P < 0.05$, LDA score>4.0). The bacteria with high relative abundances among root-associated compartments exhibited remarkable differences. A histogram of the LDA value distribution showed that 36 taxa were enriched including seven genera (*Vibrio*, *Marinomonas*, an uncultured bacterium of the family Enterobacteriaceae, *Escherichia Shigella*, *Shewanella*,

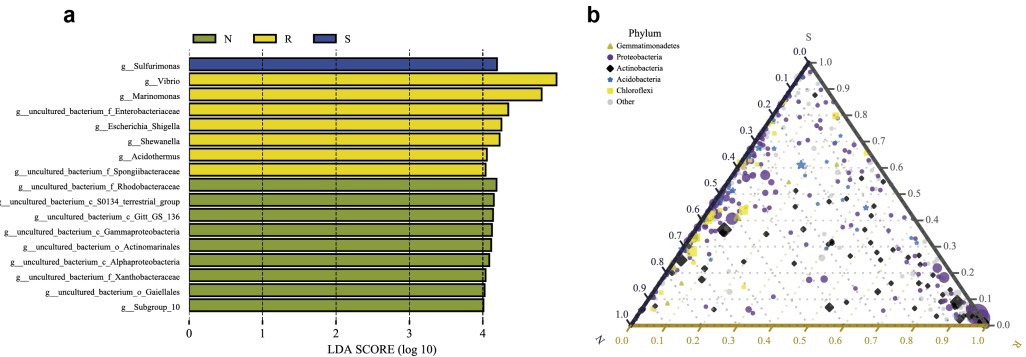

**Figure 8** **Combination of LEfSe analysis and correlation analysis.** (A) Histogram of differentially abundant genera between different compartments (logarithmic LDA score ≥ 4.0 and P ≤ 0.05). A longer bar indicates a more significant difference. The bars were colored according to the group with the highest abundance of the corresponding feature. (B) Ternary plot of the three root-associated compartments (R, S and N). The three angles of the triangle represent three compartments, they are in three colors. Three edges are used to measure the phyla richness of the compartment in corresponding colors. Different shapes with different colors represent different phyla with top five abundance, and the size of the shapes represents the average relative richness of phyla.

*Acidothermus*, and an uncultured bacterium of the family Spongiibacteraceae) in the endosphere compartment, 20 taxa were enriched including one genus (*Sulfurimonas*) in the rhizosphere compartment, while forty-one taxa were enriched including nine genera (an uncultured bacterium of the family Rhodobacteraceae, an uncultured bacterium of S0134_terrestrial_group, an uncultured bacterium of Gitt_GS_136, an uncultured bacterium of the class Gammaproteobacteria, an uncultured bacterium of the order Actinomarinales, an uncultured bacterium of the class Alphaproteobacteria, an uncultured bacterium of the family Xanthobacteraceae, an uncultured bacterium of the order Gaiellales, and Subgroup 10) in the non-rhizosphere compartment (Fig. 8A and Fig. S3). The result indicated these genera enriched in each compartment could be used as biomarkers. In the four mangrove species, two biomarkers were identified in the samples of *A. ilicifolius*, eighteen biomarkers were found in the samples of B. gymnorrhiza, one biomarker was identified in the samples of *C. inerme*, while twelve biomarkers were found in the samples of *L. racemosa* (Fig. S3).

The ternary plot uses an equilateral triangle to describe the relationship among the ratio of attributes of three rhizocompartment. In our study, a ternary plot was used to compare the richness of the top five phyla in three different root-associated compartments (endosphere, rhizosphere, and non-rhizosphere). As shown in Fig. 8B, the top five phyla are widely distributed in all three compartments, but the relative abundance was quite different. The endosphere compartment (R) had the highest abundance of Proteobacteria and Actinobacteria compared to the other two compartments whereas the distribution of the top five phyla between the rhizosphere (S) and non-rhizosphere (N) compartments were more similar. These results were consistent with the composition analysis of microbial communities discussed above.

## Correlation network analysis of bacterial communities between mangrove species

The co-occurrence networks of the endosphere (R) and rhizosphere (S) compartments were analyzed by the Pearson algorithm. As shown in Fig. 9A, the endosphere compartment showed a completely different network structure compared to the rhizosphere compartment. Among the top ten most abundant genera, the network in the endosphere compartment consists of 46 edges. An uncultured bacterium of the family Moraxellaceae, an uncultured bacterium of the order Alteromonadales, an uncultured bacterium of the family Enterobacteriaceae, and *Acidothermus* were found to be the hub genera ($\geq 6$ edges per node) in the network in which an uncultured bacterium of the family Enterobacteriaceae and *Acidothermus* were the biomarker taxa. *Vibrio*, the most dominant genus in the samples, and also one of the biomarker taxa, had a negative relationship with *Acidothermus* and *Mycobacterium* but had a positive relationship with *Shewanella*. In the rhizosphere compartment, the network structure with more edges was more complicated compared to the endosphere compartment. As the most dominant genus and the only biomarker genus, *Sulfurimonas* had the highest degree of connections (13 edges). It had a positive relationship with seven genera and a negative relationship with six genera (Fig. 9B). Interactions between microbes have a great impact on the growth and production of metabolites, therefore, further validation is needed in future studies.

## 16S functional genes prediction of bacterial communities of root-associated compartments

In our study, functional prediction of the top 25 biological functions of the bacterial populations was performed using the FAPROTAX dataset (Figs. 10A–10D). Among different mangrove species, nitrate reduction in the endosphere was much stronger than that in the other two compartments. This result was consistent with the differences in the bacterial community since the genera *Vibrio*, *Marinomonas*, and unclassified Enterobacteriaceae were involved in the function of nitrate reduction (Fig. 10A). In the root endosphere of *A. ilicifolius* and *L. racemosa*, nitrogen and nitrate respiration had a higher proportion than that of other compartments, which was consistent with the corresponding functions of genera *Marinomonas* and unclassified Enterobacteriaceae (Fig. 10B). Similarly, in the non-rhizosphere and rhizosphere compartments, the function of photoautotrophy was generally higher than that of the endosphere compartment, while the function of photoautotrophy with a great proportion in all the compartments of *C. inerme*, which was also related to the microbial composition, as these compartments were rich in a high percentage of related bacteria, *i.e.,* Chloroflexi (Figs. 10B–10C).

## DISCUSSION

Understanding the diversity and structure of root-associated bacterial microbiomes in different mangrove tree species holds crucial importance for harnessing microbial resources and comprehending their ecological functions within tropical mangrove ecosystems. In our present work, we verified the distinct diversity of bacterial communities among the three rhizocompartments across the four mangrove species. Notably, the bacterial

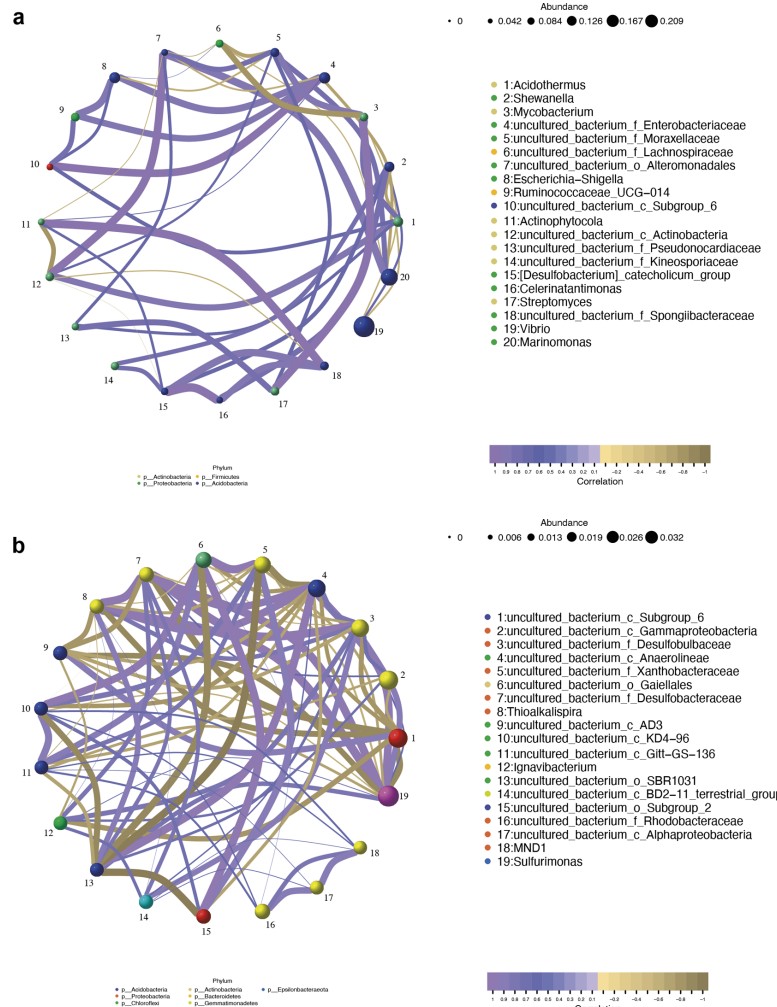

**Figure 9 Network analysis of top 20 bacterial genera in the endosphere (A) and rhizosphere (B) compartments.** The nodes were colored according to different modularity classes. The size of each node was proportional to the number of connections. Positive correlations were displayed in violet and negative correlations were displayed in olive yellow. A connection represented a strong (Pearson's correlation coefficient $|r| > 0.9$) and significant ($p < 0.05$) correlation.

communities exhibited marked diversity disparities between the endosphere and the other two compartments, irrespective of the specific mangrove species. At the phylum level, considerable overlap in the bacterial community structure was observed within the same rhizocompartment across diverse mangrove species. However, notable dissimilarities were evident at the genus level. Furthermore, we report, for the first time, the presence of *Acidothermus* within mangrove environments. Overall, our results substantially support the fundamental hypothesis that microbiome composition exhibits variations across compartments and host mangrove species.

Because culture-independent or traditionally used molecular methods inevitably resulted in an underestimation of microbial diversity, high-throughput sequencing methods are now

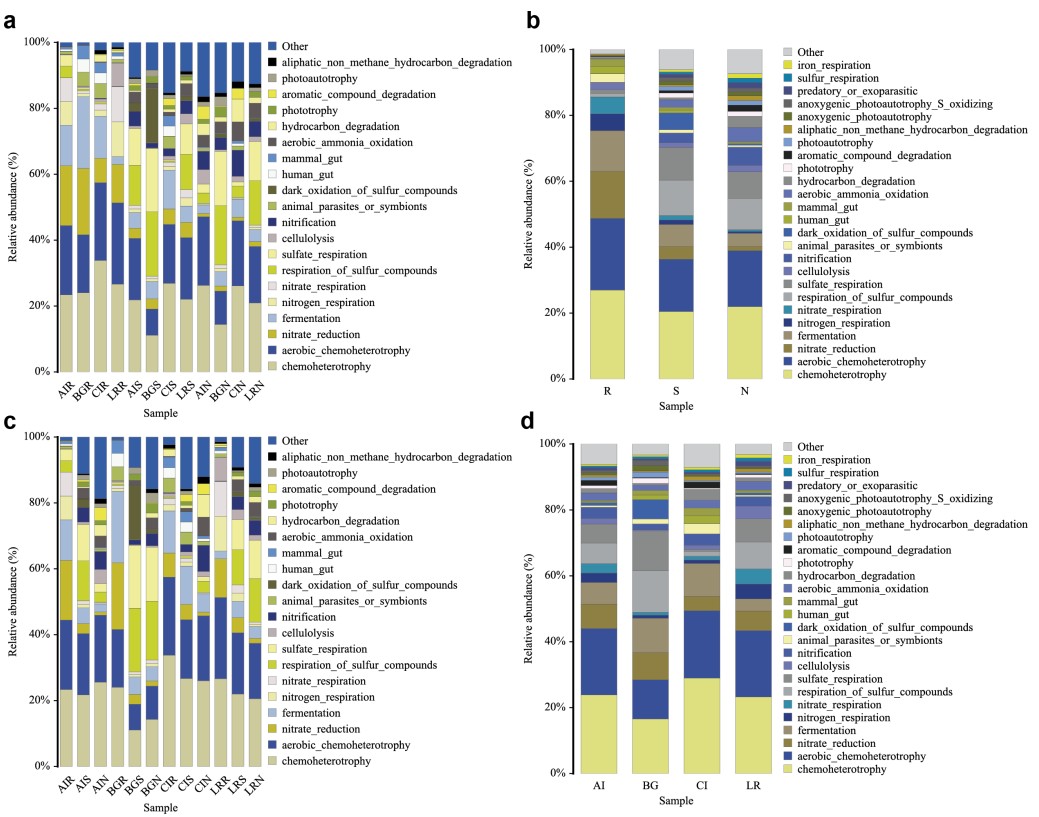

**Figure 10** Functional prediction of root-associated compartments in different mangrove species by FAPROTAX.

extensively used for investigating bacterial community structures in mangroves (*Alzubaidy et al., 2016*; *Andreote et al., 2012*; *dos Santos et al., 2011*; *Gomes et al., 2010*; *Hong et al., 2015*; *Sanka Loganathachetti, Poosakkannu & Muthuraman, 2017*; *Simoes et al., 2015*; *Wu et al., 2016*; *Zhang et al., 2017*; *Zhuang et al., 2020*). Root-associated symbiotic or epiphytic microbiomes of mangrove trees play key roles in mediating the growth and development of the host plants, the synthesis, and the accumulation of host secondary metabolites, and hence have captured more and more attention by microbiologists and pharmaceutical specialists. Nowadays, an increasing number of studies have invested the microbial diversity and its variations among distinct root-associated compartments of different mangrove trees, such as *Avicennia marina*, *Rhizophora mangle*, and *Kandelia obovata* (*Gomes et al., 2010*; *Hong et al., 2015*; *Sanka Loganathachetti, Poosakkannu & Muthuraman, 2017*; *Simoes et al., 2015*; *Zhuang et al., 2020*). However, the comprehensive understanding of microbiomes in distinct mangrove species from the identical mangrove ecosystem is still poor, especially the bacterial community composition. In our study, we determined and analyzed the bacterial community structures and diversity in different root-associated compartments across four different mangrove species *via* high-throughput sequencing.

At different taxonomic levels, the endosphere and the other two compartments exhibited distinctly diverse communities, while the bacterial communities of the rhizosphere and

non-rhizosphere compartments greatly overlapped. The bacterial community analysis revealed that phylum Proteobacteria was the most common bacteria in all root-associated compartments of the four mangrove trees. Proteobacteria with high diversity extensively existed in marine environments, and are famous for participating in the process of nutrient cycling, such as phototrophs, autotrophs, and heterotrophs (*Zhou et al., 2020*). However, the second predominant phylum was different in different compartments of the four mangrove trees. In the endosphere compartment, the second predominant phylum was Actinobacteria, while Chloroflexi was the second predominant phylum in the rhizosphere and non-rhizosphere compartments (Fig. 2A). Members of the Actinobacteria phylum are best known for soils and plant rhizospheres. Actinobacteria are well-known as symbionts and pathogens in soil-associated and marine-associated microbial communities. They are well known as excellent producers of naturally derived lead compounds in clinical applications (*Barka et al., 2016*). Consequently, they are outstanding players in the field of biotechnology, medicine, and agriculture. Meanwhile, members of the phylum Chloroflexi are composed of facultatively aerobic bacteria and probably participate in the degradation of amino acids and carbohydrates. In the roots from trees of each species, except *B. gymnorrhiza*, the second predominant phylum was Actinobacteria in the other three mangrove plants. In *B. gymnorrhiza*, Firmicutes was the second predominant phylum, which may be due to the roots of *B. gymnorrhiza* having been immersed in seawater for a long time (either at high tide or low tide), and the actinobacteria cannot survive normally due to extreme hypoxia, while Firmicutes can form spores, which can withstand various extreme environments.

At the genus level, *Vibrio*, *Marinomonas,* and *Acidothermus* were the predominant genus found in the bacterial communities in the root compartments across the four mangrove species, which is quite different from rhizosphere and non-rhizosphere compartments (Fig. 3C). LEfSe analysis showed that seven genera were enriched including *Vibrio*, *Marinomonas,* and *Acidothermus* which can be used as biomarkers (Fig. 8A). The co-occurrence networks of the endosphere compartment showed that *Acidothermus* was one of the hub genera ($\geq 6$ edges per node) (Fig. 9A), while *Vibrio* had a negative relationship with *Acidothermus*. *Vibrio* and *Marinomonas* both belong to Gammaproteobacteria Class. This class includes populations that can decompose marine organic matter by nitrate reductions (*Gomes et al., 2010*). Members of the genus *Vibrio* are motile bacteria that usually existed in estuarine or marine environments. Quite a few species of *Vibrio* can interact with other organisms, which can be salutary symbionts or fatal pathogens (*DeAngelis, Saul-McBeth & Matson, 2018*). In the roots of mangrove species, it is possible that Vibrios can sense and respond to various stresses to adapt to the internal environment of mangrove plants and perform their biological functions. *Marinomonas* was another predominant genus of Gammaproteobacteria Class found in the roots of the four mangrove trees, not the other two compartments (Figs. 3C and 4D). Some members of the genus *Marinomonas* can adapt to various types of environments and perform a variety of metabolic functions (*Xue et al., 2022*). As shown in Fig. 4D, *Marinomonas* accounted for a greater proportion of the roots of mangrove *L. racemosa* and *A. ilicifolius*, whereas it was depleted in *C. inerme* and *B. gymnorrhiza*, which may be adapted to their environment. *L. racemosa* and *A. ilicifolius*

grow in semi-muddy soil, while *C. inerme* in sandy soil, *B. gymnorrhiza* in muddy soil where constant immersion in seawater leads to extreme hypoxia. *Acidothermus* is another genus of the Actinobacteria Class which is one of the hub genera and with high abundance only in the root endosphere in this study. *Acidothermus* can withstand hot and acid and has great potential in the microbial conversion of biomass. Many studies have been conducted on the cellulose decomposition activity shown by members of *Acidothermus* (*Das et al., 2020*; *Hengge et al., 2022*). This is the first report of *Acidothermus* found in mangrove environments. The results revealed that it may be of significant importance to isolate the microorganisms of this genus by pure culture method to find new hydrolytic activities or new strains can be found.

Microbial diversity and community structure in root-associated compartments were influenced by a variety of factors, including selection in the assembly of microbial communities, environmental habitat, geographical locations, host plants, and the way of preprocessing after sampling. In our study, we selected four mangrove plants with different root microenvironments in the same mangrove forest as the research objects. Here, we consider three factors that may contribute to our results: (i) spatial variation of the bacteria in different rhizocompartments affects the assembly of microbes. Similar patterns in Arabidopsis, rice, and mangrove *Kandelia obovate* microbiota revealed that Proteobacteria accounted for a higher proportion in the endophere than in the other two exterior rhizocompartments, whereas Chloroflexi and Acidobacteria were almost depleted in the endosphere (*Edwards et al., 2015*; *Lundberg et al., 2012*; *Zhuang et al., 2020*). In our study, our results also showed Proteobacteria was the most common bacteria in the endophere, whereas Chloroflexi, Acidobacteria, and the other two phyla (Gemmatimonadetes, and Bacteroidetes) were also almost depleted in the endosphere of all four mangrove species. (ii) the root microenvironments of mangrove species are closely related to microbial community structure. The diversity of bacteria was similar at the phylum level but significantly different at the genus level in the root-associated compartments in this study. Interestingly, these samples were collected in the same geographical location, we inferred there may be two reasons. On the one hand, the mangrove populations we selected were different species, so the host plants were different and produced different carbon metabolites which were received by the root-associated microbes *via* root exudates (*Sasse, Martinoia & Northen, 2018*); On the other hand, the four disparate mangrove trees grew in the different soil environment. *A. ilicifolius* we selected grew in yellow semi-muddy soil on coastal slope where seawater was flowing and was flooded or exposed by the tide, *B. gymnorrhiza* grew in black muddy soil which contained decayed plants and was soaked in seawater all year round. *L. racemosa* grew in yellow-brown semi-muddy soil, while *C. inerme* grew in sandy soil which was much like terrestrial soil and was submerged by sea water for less time. Therefore, host plants and their surrounding environment may affect the microbial community structure and diversity. Third, compared to previous studies, root-associated microbial composition from samples from different sites (Maoming, Guangdong, China; Zhangzhou, Fujian, China; Guanabara Bay, southeastern Brazil, and Haikou, Hainan, China) differed greatly, indicating that geographical location may also affect root microbiomes (*Gomes et al., 2010*; *Hong et al., 2015*; *Zhuang et al., 2020*).

However, this study has a few limitations. First, the samples of the mangrove plant species were limited, so the results of this study were not fully representative of the microbial distribution of all mangrove plants; Second, there were differences in the soil types in which the mangrove plants we selected were located. Although it has been confirmed that the assembly of root-associated microbes depends on compartment niche and host species, soil types may still be a contributing factor, thus further studies were warranted (*Muwawa et al., 2021*; *Wu et al., 2016*; *Xiong et al., 2021*); Third, The samples we selected were naturally grown, so it was difficult to find root-related samples of different host mangrove plants with exactly the same growth stage. Although we did five biological replicates, each comprising five pooled subsamples for each mangrove species, the results would still be biased to some extent. Therefore, the resulting analysis might show only a rough relationship.

## CONCLUSIONS

In our study, we comprehensively investigated the bacterial microbiome across the three root-associated compartments in four different mangrove species by high-throughput amplicon sequencing. The bacterial communities between the endosphere and the other two compartments were distinctly diverse independent of mangrove species. The bacterial community structure of the same rhizocompartment across distinct mangrove species revealed a large degree of overlap at the phylum level while it was quite different at the genus level. Our study provides unique insight into the in-depth understanding of microbial composition, diversity, and potential function across three compartments in four different mangrove species in the same mangrove ecosystem. Understanding the complex biochemical and molecular crosstalk between plants, soil, and microbes, and then exploring novel bioactive natural products would be the most important in future studies.

Regarding the study's limitations, it should be noted that our knowledge of microbiome composition and predicted functions are restricted to findings from sequencing of 16S rRNA gene amplicons. As a result, the information on microbiota composition and potential functions should be explained conditionally. In addition to identifying more novel metabolic and microbial biomarkers, future research based on metagenomic sequencing and metabolomics of the microbiota-dependent metabolites could provide the basis for protecting mangrove ecosystems and exploiting and utilizing the microbial resources. Environmental changes affect microbial diversity and thus alter the capacity of microbes to sustain ecosystem functions. Therefore, the correct monitoring of microbial population changes to address changes in mangrove ecosystems will be the direction of future research.

## ACKNOWLEDGEMENTS

We gratefully acknowledge the staff of the Dongzhaigang National Natural Reserve for their support and assistance in carrying out sampling.

### Funding

This work was supported by the Natural Science Foundation of Hainan Province (Grant No. 2019RC216), the Introduction of Talent Research Start-up Fund of Hainan Medical College (Grant No. XRC190006), and the Natural Science Foundation of Hainan Province (Grant No. 822RC708). The funders had no role in study design, data collection and analysis, decision to publish, or preparation of the manuscript.

### Grant Disclosures

The following grant information was disclosed by the authors:
Natural Science Foundation of Hainan Province: 2019RC216.
Introduction of Talent Research Start-up Fund of Hainan Medical College: XRC190006.
Natural Science Foundation of Hainan Province: 822RC708.

### Competing Interests

The authors declare there are no competing interests.

### Author Contributions

- Jinlei Sui conceived and designed the experiments, performed the experiments, analyzed the data, prepared figures and/or tables, authored or reviewed drafts of the article, and approved the final draft.
- Xiaowen He performed the experiments, prepared figures and/or tables, authored or reviewed drafts of the article, and approved the final draft.
- Guohui Yi conceived and designed the experiments, analyzed the data, authored or reviewed drafts of the article, analysis, and approved the final draft.
- Limin Zhou performed the experiments, prepared figures and/or tables, authored or reviewed drafts of the article, and approved the final draft.
- Shunqing Liu performed the experiments, analyzed the data, prepared figures and/or tables, authored or reviewed drafts of the article, and approved the final draft.
- Qianqian Chen performed the experiments, analyzed the data, authored or reviewed drafts of the article, and approved the final draft.
- Xiaohu Xiao conceived and designed the experiments, analyzed the data, prepared figures and/or tables, authored or reviewed drafts of the article, and approved the final draft.
- Jinyan Wu conceived and designed the experiments, prepared figures and/or tables, authored or reviewed drafts of the article, and approved the final draft.

### DNA Deposition

The following information was supplied regarding the deposition of DNA sequences:
The sequences are available at Genbank: PRJNA951277.

### Data Availability

The sequences are available at Genbank: PRJNA951277.
https://www.ncbi.nlm.nih.gov/bioproject/PRJNA951277/

## Supplemental Information

Supplemental information for this article can be found online at http://dx.doi.org/10.7717/peerj.16156#supplemental-information.

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
