# Peer review of "Diversity and structure of the root-associated bacterial microbiomes of four mangrove tree species, revealed by high-throughput sequencing"

_PeerJ, doi:10.7717/peerj.16156_

## Round 0.1 · original submission · Major Revisions

The reviewers have gone through the manuscript and made series of comments on the manuscript. Please refer to these comments and revise the manuscript accordingly. A point-by-point response letter is needed for the resubmission.

·

Basic reporting

In my opinion, this is a well-written, updated, and clear work. dealing with comprehensively investigating the bacterial microbiome across the three root-associated compartments in four different mangrove species by high-throughput amplicon sequencing. The significant overlap in key community members of the same rhizocompartment among distinct mangrove species was found at the phylum level. Furthermore, this is the first report of Acidothermus found in mangrove environments.
▒ A Clear, unambiguous, professional English language is used throughout the whole manuscript a few language and punctuation problems in the whole text need editing which were listed below:
Line 47: remove the comma.
Line 108: add missing verb ( the samples were divided).
Line 111: Fix the agreement mistake (The soils) and add the missing verb (is attached)
Line 120: remove the verb (were) in were used for DNA isolation
Line 136: correct article usage base-calling to be a base-calling
Line 150: add the passive form Simpson and Shannon's
Line 160: correct article usage The FAPROTAX
Line 172: add a comma (, and the soil)
Line 173: correct article usage (of the other)
Line 174: correct the quantifier than to be ( more than)
Line 177: change the capitalization shannon to be ( Shannon)
Line 199: add an article proportion to be (the proportion)
line 216: correct article usage root to be the root
line 240: change the capitalization simpson to be Simpson
line 241: change the verb form two compartments was to be (two compartments were)
line 250: correct article usage Figure (the Figure)
line 271 : change the preposition on to be( in different compartments)
line 284, 289,290,291, 292 and 312: add an article uncultured to be (an uncultured)
line 305: change the verb form compartments was to be were
line 337: correct your spelling i.e Chloroflexi to be (i.e. Chloroflexi)
line 352: correct subject-verb agreement is to be (are still poor) and remove the preposition for
line 374: change the verb form was to be were
line 395: change the preposition in to be of
line 399: correct article usage genus of Actinobacteria to be (genus of the Actinobacteria)
line 428: add an articles coastal slope to be (the coastal slope)
line 453 : add comma and then exploring to be (, and then exploring)

▒ Abstract: need to write more detail about high-throughput sequencing and clarify the type of PCR used to detect the results whether it is conventional or nested or real-time.
▒ The intro & background are well-referenced, and relevant, and covers the subject of the publication.
▒ The structure conforms to PeerJ standards, discipline norm
▒ Figures are relevant, high quality, well labeled & described.
▒ Raw data supplied according to Peer J policy

Experimental design

▒Experimental design Original primary research within the scope of the journal (applicability to the core areas of biological and environmental sciences) and included bioinformatics software analysis.
The research question is well defined, relevant, and meaningful.
Rigorous investigation performed to a high technical and ethical standard
Methods are described with sufficient detail and information to replicate them.
Discussion: This section is well presented.
References: is well presented and follows a consistent pattern.

Validity of the findings

▒ Impact and novelty are assessed. Meaningful replication is encouraged where the rationale and benefit to literature are clearly stated.
▒ All underlying data have been provided; they are robust, statistically sound, and controlled.
▒ Conclusions are well stated, linked to the original research question, and limited to supporting results.

·

Basic reporting

The manuscript presents a comprehensive study focusing on the bacterial communities found in the roots of four distinct mangrove tree species, namely Acanthus ilicifolius, Bruguiera gymnorrhiza, Clerodendrum inerme, and Lumnitzera racemosa. The research findings significantly contribute to our understanding of the intricate relationships between mangrove species and the microbes associated with them. Moreover, these findings have important implications for harnessing microbial resources and exploring the production of secondary metabolites. Although the manuscript is well-written, it is crucial to mention and provide information about the availability of associated data. Therefore, I strongly recommend including details about data availability in the manuscript.

Experimental design

The Materials and Methods section of the manuscript provides a comprehensive description of the sampling procedures, DNA extraction, PCR amplification, high-throughput sequencing, and subsequent data analysis. However, there are a few areas where further clarification and expansion would enhance the clarity and reproducibility of the experimental procedures:

It is recommended to include the specific date or time period when the sampling was conducted in the summer of 2021. This information will provide a clearer context for the study.

The description of the different compartments (endosphere, rhizosphere soil, and non-rhizosphere soil) could be improved by providing a brief explanation of each compartment and its relevance to the study. This will help readers understand the significance of examining these specific compartments.

In the DNA extraction process, while it is mentioned that the MoBio PowerSoilÆ DNA isolation kit was used, it would be beneficial to provide specific details or a step-by-step protocol followed during the DNA extraction. This will ensure clarity and reproducibility.

The primer pairs used for amplifying the 16S rRNA genes (V3 & V4 region) are referenced as described previously (Wang et al. 2017). To facilitate reproducibility, it is recommended to include the primer sequences or additional details about their source, such as the primer sequences used in the study.

In the bioinformatics analysis, it would be helpful to explain the specific criteria used for filtering and generating valid amplicon sequences. For example, details about the quality thresholds applied in Trimmomatic, Usearch, and UCHIME would provide clarity on the sequence filtering process.

While the general steps of the bioinformatics analysis are described, it would be beneficial to provide more specific details about the parameters and software versions used in each step. This is particularly important for data preprocessing, taxonomic annotation, and statistical analyses. Including this information will facilitate reproducibility and allow readers to understand the analysis in more detail.

The mention of the FAPROTAX database as a tool for predicting possible functions based on the 16S rRNA gene amplicon data is valuable. However, it would be beneficial to elaborate on how this database was specifically used and how the functional predictions were integrated into the study. Providing more information on this aspect will enhance the understanding of the functional analysis conducted.

By addressing these points, the clarity and reproducibility of the experimental procedures in the Materials and Methods section can be significantly improved.

Validity of the findings

The results section provides a comprehensive analysis of the sequencing data and describes the composition and diversity of bacterial communities in different root-associated compartments and mangrove species.

The discussion provides a thorough analysis of the findings and effectively relates them to previous studies. The authors offer valuable insights into the factors contributing to microbial diversity in mangrove root-associated compartments and raise important questions for further research. However, it would be beneficial to include a brief summary of the main findings and their implications at the beginning of the discussion section to provide better context for the subsequent analysis and interpretation of the results.


Overall, the conclusion effectively summarizes the key findings and their significance but could benefit from a more focused and forward-looking perspective.

Additional comments

General Comments: The introduction provides a comprehensive overview of the importance of mangrove forests, particularly the Dongzhaigang mangrove wetland in Hainan, China. It highlights the rich biodiversity, medicinal value, and unique environmental conditions of mangrove ecosystems. The manuscript aims to investigate the root-associated microbial communities of four different mangrove tree species and their potential functions. Overall, the introduction is well-written and informative. However, there are a few areas that require attention and clarification.
Specific Comments:
Clarity and Organization:
The introduction could benefit from better organization and logical flow. Consider grouping the information more coherently to guide readers through the different aspects of the study.
Clarify the specific research objectives and questions that the study aims to address. This will help readers understand the purpose and scope of the investigation.
Citations:
it would be helpful to include some direct references for specific statements made to support the claims and statements made.
Terminology and Definitions:
Define important terms and concepts such as "endophytic bacteria," "rhizosphere," and "rhizocompartments" for readers who may not be familiar with these terms.
It would be beneficial to clarify the distinction between endophytic and rhizosphere bacteria and their roles in the study.
Research Gap and Significance:
The introduction briefly mentions that "little is known about the structure of mangrove rhizomes in different compartments of different mangrove species." Elaborate on this research gap and emphasize the novelty and significance of the study in filling this knowledge gap.
Research Scope and Limitations:
Clearly state the specific mangrove species chosen for the study and explain the rationale for their selection.
Discuss any limitations or potential challenges that may arise during the investigation, such as sample collection difficulties or potential biases in the analysis.
Aim and Hypothesis:
Explicitly state the research aim and specific hypotheses that the study intends to test. This will provide a clear focus for the subsequent sections of the manuscript.
Language and Grammar:
Proofread the introduction to correct minor grammatical errors, typos, and inconsistencies in sentence structure.
On lines 56 and 57, the authors mistakenly used the abbreviated names for the trees instead of their full scientific names, which should have been provided initially and then followed by the abbreviated versions.

Overall, the introduction provides a solid foundation for the study, highlighting the importance of mangrove ecosystems and the potential role of root-associated microbial communities. Addressing the points mentioned above will enhance the clarity and effectiveness of the introduction, setting the stage for the subsequent sections of the manuscript.

---

## Round 0.2 · accepted · Accept

Based on the reviewers' comments, I recommend the acceptance of the manuscript.

·

Basic reporting

▪ Clear and unambiguous, professional English is used throughout.
▪ Literature references and sufficient field background or context are provided.
▪ Professional article structure, figures, and tables Raw data was shared.
▪ Self-contained with relevant results for hypotheses.

Experimental design

▪ Original primary research within the aims and Scope of the journal
▪ The research question is well defined, relevant, and meaningful. It is stated that research fills an identified knowledge gap.
▪ Rigorous investigation performed to a high technical and ethical standard
▪ Methods are described with sufficient detail and information to replicate them.

Validity of the findings

▪ Impact and novelty were not assessed. Meaningful replication is encouraged where the rationale and benefit to literature are clearly stated.
▪ All underlying data have been provided; they are robust, statistically sound, and controlled.
▪ Conclusions are well stated, linked to the original research question, and limited to supporting results.

·

Basic reporting

Thank you authors for working on my suggestion and for the improved manuscript.

Experimental design

No comment.

Validity of the findings

No comment.

Additional comments

No comment.